# Immunolocalization of Nesfatin-1 in the Gastrointestinal Tract of the Common Bottlenose Dolphin *Tursiops truncatus*

**DOI:** 10.3390/ani12162148

**Published:** 2022-08-22

**Authors:** Elena De Felice, Claudia Gatta, Daniela Giaquinto, Federica Fioretto, Lucianna Maruccio, Danila d’Angelo, Paola Scocco, Paolo de Girolamo, Livia D’Angelo

**Affiliations:** 1School of Biosciences and Veterinary Medicine, University of Camerino, Via Gentile III da Varano, 62032 Camerino, Italy; 2Department of Veterinary Medicine and Animal Production, University of Naples Federico II, Via F. Delpino 1, 80137 Naples, Italy

**Keywords:** Nesf-1, digestive tract, bottlenose dolphin, immunohistochemistry

## Abstract

**Simple Summary:**

Nesfatin-1 (Nesf-1) is a neuropeptide that plays important roles in regulating food intake, mainly related to its anorexigenic effect, and it is mainly distributed in the digestive systems of all vertebrates. With this study, we expand knowledge on the localization of Nesf-1 in the digestive tract of an aquatic mammalian species, the common bottlenose dolphin (*Tursiops truncatus*), allowing comparative study on terrestrial mammals. Dolphin tissue samples (three gastric chambers and intestine) were provided by the Mediterranean Marine Mammal Tissue Bank of the Department of Comparative Biomedicine and Food Science of the University of Padova (Italy).

**Abstract:**

First identified as an anorexigenic peptide, in the last decades, several studies have suggested that Nesfatin-1 (Nesf-1) is a pleiotropic hormone implicated in numerous regulatory processes in peripheral organs and tissues. In vertebrates, Nesf-1 is indeed expressed in the central nervous system and peripheral organs. In this study, we characterized the pattern of Nesf-1 distribution within the digestive tract of the common bottlenose dolphin (*Tursiops truncatus*), composed of three gastric chambers and an intestine without a clear subdivision in the small and large intestine, also lacking a caecum. Our results indicated that Nesf-1 is widely distributed in cells of the mucosal epithelium of the gastric chambers. Most of the immunoreactivity was observed in the second chamber, compared to the first and third chambers. Immunopositivity was also found in nerve fibers and neurons, scattered or/and clustered in ganglion structures along all the examined gastrointestinal tracts. These observations add new data on the highly conserved role of Nesf-1 in the mammalian digestive system.

## 1. Introduction

Feeding behavior is regulated by the complex interaction of several molecules [1]. Nesfatin–1 (Nesf-1) is a polypeptide 82 amino acids in length derived from calcium and DNA-binding protein NUCB2 (NEFA/nucleobindin-2), and it is associated to the control of food intake. Nesf-1 has three parts: the N-terminal (N23), middle (M30) and C-terminal (C29) parts. Among these, M30 represents the active part of the protein, having an anorexic effect [2]. About 85% of its structure is conserved in mammals, including humans [3]. In the central nervous system (CNS), Nucb2 mRNA is localized in several hypothalamic nuclei implicated in the regulation of feeding behavior [4,5]. Cells expressing Nesf-1 are often found to express other bioactive molecules. Nesf-1 is co-localized with ADH (antidiuretic hormone) or CRH (corticotropin) or TRH (thyrotropin-releasing hormone) in hypothalamic neurons [6]. Nesf-1 is thus a key regulator of alimentary requirements and principally acts via the anorexigenic system of the melanocortin, independent of the leptin pathway. Intracerebroventricular administration of Nesf-1 and NUCB2 decreases food intake and body weight while increasing sympathetic nerve activity and blood pressure [4]. Outside the CNS, Nucb2 mRNA is mostly localized in the pancreas, gastric mucosa, duodenum and white adipose tissue [7,8,9,10] and, further, but to a lesser extent, in other peripheral organs, e.g., testis [11]. Very interestingly, Stengel and co-workers demonstrated that NUCB2/Nesf-1 expression levels in endocrine cells of gastric mucosa were 10-fold higher compared to brain levels, supporting the idea that the stomach is one of the main sites of production of circulating NUCB2/Nesf-1 [8,12]. Like in mammals, in teleost fish, Nesf-1 was expressed in both central and peripheral tissues, and different studies have demonstrated the highest NUCB2/Nesf-1 mRNA expression in the gastrointestinal tract [13,14,15], hypothesizing a common role of this peptide in vertebrates.

In vertebrates, the evolution of the gastrointestinal tract is correlated with specific metabolic needs and individual demands for processing, distributing and adsorbing nutrients, and it ultimately modulates body weight [16]. The common bottlenose dolphin *Tursiops truncatus* shows a diverticulated composite stomach, consisting of three chambers: the forestomach (or first chamber), main stomach (or second chamber) and pyloric stomach (or third chamber) [17]. With regards to intestine, there is not a clear subdivision into small and large intestine [18]. The presence of a multiple-chambered stomach, uncommon in carnivores, is due to the fact that dolphins are phylogenetically related to arctiodactyls, which also have multichambered stomachs [19].

Based on the key morpho-physiological features of the gastroenteric tract of the common bottlenose dolphin, we undertook this study to explore the pattern of distribution of Nesf-1 to conduct further comparative studies with other mammals, terrestrial and marine [17,20,21].

## 2. Materials and Methods

### 2.1. Animals and Sample Preparations

To conduct this study, we used a series of samples of the alimentary canal from 3 adult males of *Tursiops truncatus* (common bottlenose dolphin) kindly furnished by the Mediterranean Marine Mammal Tissue Bank (MMMTB) of the Department of Comparative Biomedicine and Food Science of the University of Padova (Italy). Stored tissues in the MMMTB (CITES institution IT020) derived from stranded animals or from marine mammals who died in captivity and were referred for postmortem. Therefore, ethical approval was not required for this study. The samples analyzed were the following: first chamber or forestomach, second chamber or main stomach, third chamber or pyloric stomach and intestine. Organ tissue sampling was carried out within a few hours after death; samples were treated for fixation with 10% buffered formalin and then embedded in paraffin wax for histochemical purposes.

### 2.2. Single Immunohistochemistry

The distribution of Nesf-1 was studied by immunohistochemistry. To obtain 8 µm thick sections, paraffin-embedded samples were serially and transversally cut. In order to block the endogenous pseudo-peroxidase activity, the dewaxed sections were incubated with 3% hydrogen peroxide for 30 min at room temperature (RT), rinsed for 15 min in 0.01 M phosphate-buffered saline (PBS) pH 7.4 and then incubated for 20 min at RT with normal goat serum (NGS, 1:5 in 0.01 M PBS) (MP biomedicals LLC, cat# 191356) in order to prevent background signal noise. The sections were then incubated in a humid chamber overnight at 4 °C with polyclonal antibody against Nesf-1, N-terminal (human) raised in rabbit (1:500) (H-003-24; Phoenix Pharmaceuticals, Belmont, CA, USA). The next day, the sections, after rinsing in PBS for 15 min at RT, were incubated for 30 min at RT with Immuno Reagents, Inc (cat# UNIHRP-015). A solution of 10 mg 3–3′ diaminobenzidine tetrahydrochloride (DAB, Sigma–Aldrich cat# D5905, St. Louis, MO, USA) in 15 mL of 0.5 M Tris buffer, pH 7.6, containing 0.03% hydrogen peroxide was used to detect the immunoreactive sites.

### 2.3. Controls of Specificity

Samples from common bottlenose dolphin pancreas were used as positive controls [21]. Negative controls were carried out by substituting primary antisera or secondary antisera with PBS or normal serum in the respective steps. Control images are reported in Appendix A.

### 2.4. Image Acquisition

Light images were observed and analyzed using a Nikon Eclipse 90i. The digital raw images were optimized for image resolution, contrast, evenness of illumination and background using Adobe Photoshop (Adobe Systems, San Jose, CA, USA).

## 3. Results

The gastroenteric tract of *Tursiops truncatus* is composed of three gastric chambers (Appendix A): the forestomach or first chamber, main stomach or second chamber, and pyloric stomach or third chamber. The duodenal ampulla is just an initial enlargement of the intestine. Macroscopic subdivisions into small and large intestine are not distinguishable [17,18].

Nesf-1 was distributed over the entire wall of the first and second chambers, with the highest intensity in single nerve fibers or organized in small bundles (Figure 1a,b). Nerve fiber immunoreactivity was observed in the perivascular areas of small arteries and arterioles in both the first and second chambers (Figure 1a,b). Positive immunoreactivity was also observed in neurons of the myenteric plexus, more intensely stained in the second chamber (Figure 1c). Still in the second chamber, wider immunoreactivity was detected in cells of the mucosa (Figure 1d,e) and small scattering fibers in the submucosa layer (Figure 1d).

In the third chamber, immunoreactive (ir) nerve fibers were found scattered and clustered in the myenteric plexuses (Figure 2a–c). Many strongly immunostained neurons were found scattered or isolated (Figure 2c), but most often they were clustered and belonged to ganglion structures (Figure 2a,b). The presence of few Nesf-1 ir-cells was detected in the mucosal epithelium of the third chamber (Figure 2d).

Immunopositive neurons either isolated (Figure 3a) or clustered in ganglia structures (Figure 3b) were scattered throughout the intestine. Many fibers were observed in the intestinal tract (Figure 3a–c). No positive cells were detected in the mucosal epithelium of the gut.

Table 1 recapitulates the positive cells, neurons and fiber localization of Nesf-1 protein in the gastrointestinal tract of *T. truncatus*. Positive cells in the mucosa were detected in the second and third chambers, while they were absent in the first chamber and intestine. The number of neurons identified was greater in the third chamber and decreased in the second chamber and in the intestine. Fibers were present in greater quantities along the entire wall of the first and third chambers, while they were gradually reduced in the second chamber and intestine.

## 4. Discussion

The common bottlenose dolphin has a three-compartment stomach. The first chamber or forestomach is characterized by the presence of folds capable of spreading out to favor the transit of food. The outer layer of the epithelial tissue is almost completely covered by keratinized stratified squamous epithelium and dendritic surface cells [22], while glands are absent [23]. The second chamber or main stomach is considered homologous to the gastric fundus of terrestrial mammals. The lamina propria of the mucosa is filled with tubular gastric glands, while the *muscularis mucosae* is discontinuous and is composed of two layers [24]. The third chamber or pyloric stomach has a relatively simple structure, showing a mucosa with tubular glands and an external musculature thinner than that of the more anterior compartments [25]. The complexity of this organization, in particular, the presence of a large forestomach, is probably due to the fact that dolphins do not chew their food and there is a need to grind the ingested fish or squid. Besides the difficulty in appreciating macroscopic subdivisions into small and large intestine, it is noteworthy that this species lacks a caecum. Furthermore, at the microscopic level, there are no marked differences between the anterior and posterior parts of the intestine [18], as it appears as a long unvarying tube of homogenous caliber and relatively constant aspect [20]. The mucosa, like that in terrestrial mammals, shows the typical epithelium with absorptive cells and relevant folds, lying on a connective lamina propria showing small lymphocytes, plasma cells and blood capillaries [26]. Harrison and co-workers [27] described three differently located nervous plexuses that together constitute the enteric nervous system (ENS). The first is the mucosal plexus at the *muscularis mucosae* level, the second corresponds to the submucosal plexus (Meissner’s plexus) on the internal surface of the *muscularis externa*, and the last is represented by the myenteric plexus (Auerbach’s plexus) between the layers of the *muscularis externa*.

The distribution pattern of Nesf-1 is here described for the first time in the digestive apparatus of the common bottlenose dolphin, exhibiting a peculiar distribution when compared to the terrestrial mammals, despite the 95% homology of the amino acid sequences between the *Tursiops truncatus* and human Nesf-1 proteins [21]. While we observed faint immunopositivity only in nervous fibers of the ENS in the first chamber, we appreciated Nesf-1-positive cells in the epithelium of mucosal glands in both the second and third chambers. The absence of immunopositivity in the epithelium of the first chamber is coherent with the supposed mechanic function of food squeezing of this gastric chamber, without any absorption and/or neuroendocrine function. Very interestingly, we observed intense staining in the myenteric intestinal ganglia, as well as in neuronal fibers, while the absorptive epithelium was devoid. In both rodents and humans, NUCB2/Nesf-1 has been revealed in the middle and lower portions of gastric mucosal glands [8] and in the submucosal layer of the duodenum [28]. It would be interesting to compare data on Nesf-1 in terrestrial ruminants, the phylogenetically related species to *Tursiops truncatus*, to allow more in-depth evolutionary analyses on the potential role of NUCB2/Nesf-1 in the digestive apparatus. Of interest, we observed the highest immunopositivity in the neurons and nerve fibers of all examined tracts, suggesting a role for this peptide in the regulation of secretive and motility activities of the gastrointestinal tract, as occurs in terrestrial mammals. Varricchio and co-workers [29] observed Nesf-1 immunoreactive neurons and nerve fibers in the duodenal internal submucous plexus, in the external submucous plexus of the ileum and in the myenteric plexus of the colon in pig. The ENS exerts different functions, like the control and coordination of local motility, the flow of fluids through the mucous epithelium, regulation of blood flow and interplay with the immune system [30]. It has been demonstrated that both peripheral and central administration of Nesf-1 in rats inhibits gastric acid and pepsin secretion and induces hyperemia, thus activating a protective mechanism resulting in an empowerment of the gastric mucosal defense system [31], while intracerebroventricular injection of Nesf-1 in mice exerted an inhibitory action on antral and duodenal motility [32]. In addition, peripheral injection of Nesf-1 decreased gastric contractions and halted cyclical interdigestive migrating contractions in fasted Beagle dogs [33]. Further experimental approaches could be essential to disentangle the role of NUCB2/Nesf-1 on the ENS of bottlenose dolphins and to extrapolate data on cetaceans. However, this is not feasible in *T. truncatus*, since it cannot be used as an experimental animal. Experimental observations, as in the case of this study, are limited to stranded wild specimens and, only occasionally, captive animals; this represents a limitation in conducting studies on this species.

## 5. Conclusions

Our report adds novel information on the distribution of Nesfatin-1 in the gastrointestinal tract of dolphins, in light of the evolutionary position of these mammalian species. While some shared features between common bottlenose dolphins and terrestrial mammals are conserved, others prompt us to consider a species-specific adaptation. As common bottlenose dolphins are cetaceans highly nested within the artiodactyl phylogenetic tree, it will be interesting to form a comparison with terrestrial ruminants in further research to fully understand the intricate interplay among evolution, feeding habits and digestive physiology.

## Figures and Tables

**Figure 1 animals-12-02148-f001:**
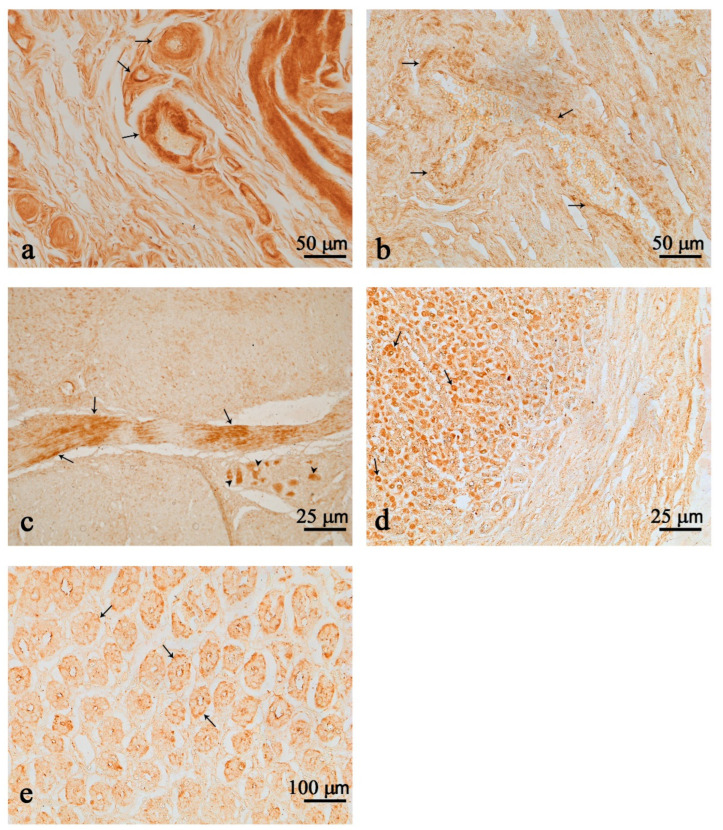
Nesfatin-1 (Nesf-1) immunoreactivity in the first (**a**) and second (**b**,**e**) chambers. Nerve fibers (arrows) in perivascular locations in the first (**a**) and second (**b**) chambers. Nesf-1 positivity in numerous fibers (arrows) and few clustered immunopositive neurons of the second chamber (**c**) (arrow heads). Positive cells in the mucosal epithelium of the second chamber (arrows) (**d**,**e**). Positive fibers in the submucosal layer (**d**). Scale bar: (**a**,**b**) = 50 µm; (**c**,**d**) = 25 µm; (**e**) = 100 µm.

**Figure 2 animals-12-02148-f002:**
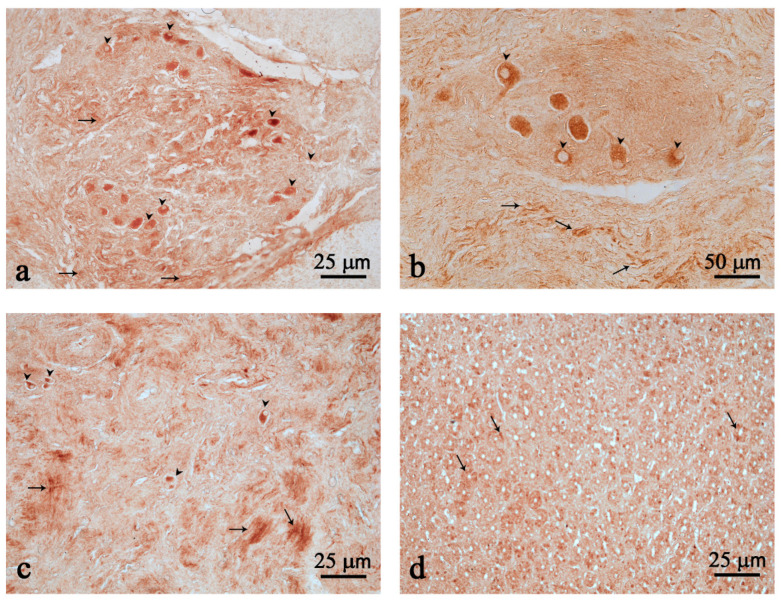
Nesfatin-1 (Nesf-1) immunoreactivity in the third chamber. Nesf-1 distribution in ganglion structures (**a**,**b**); positive fibers (arrows) (**a**–**c**); scattered neurons (arrow heads) (**a**–**c**); mucosal epithelium (arrows) (**d**). Scale bar: (**a**,**c**,**d**) = 25 µm; (**b**) = 50 µm.

**Figure 3 animals-12-02148-f003:**
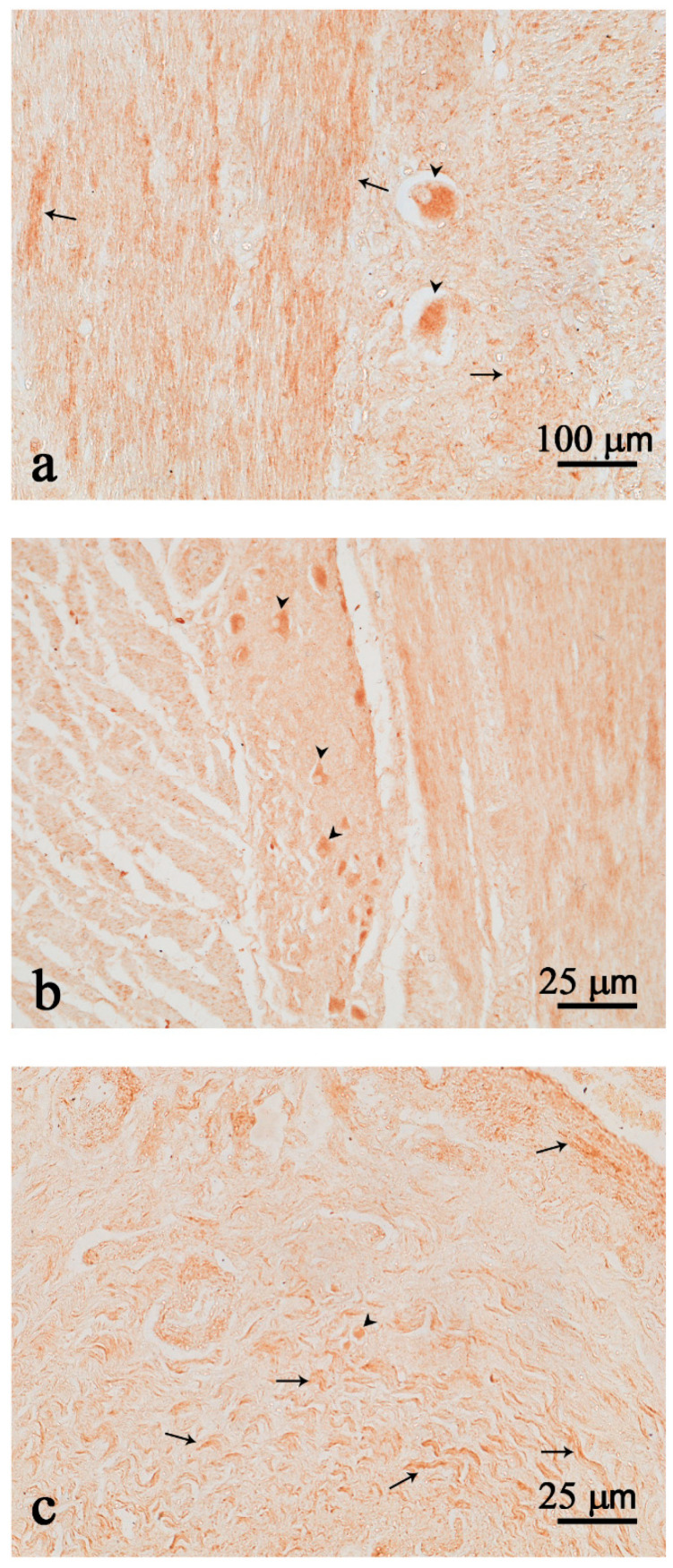
Nesfatin-1 (Nesf-1) immunoreactivity in the intestine. Nesf-1 positivity is evident in few isolated neurons (**a**,**c**); neurons in the myenteric ganglion (**b**); nerve fibers (arrows) (**a**–**c**); neurons (arrow heads) (**a**–**c**). Scale bar: (**a**) = 100 µm; (**b**,**c**) = 25 µm.

**Table 1 animals-12-02148-t001:** Summary of immunolocalization of Nesf-1 in the gastrointestinal tract of common bottlenose dolphin.

	Mucosal Epithelium	Enteric Nervous System	
		Neurons	Fibers
First chamber	−	−	+
Second chamber	+	+	+
Third chamber	+	+	+
Intestine	−	+	+

## Data Availability

All data generated or analyzed during this study are included in this published article.

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
