# Peer review of "Immunolocalization of Nesfatin-1 in the Gastrointestinal Tract of the Common Bottlenose Dolphin Tursiops truncatus"

_animals, 2022, doi:10.3390/ani12162148_

Round 1
Reviewer 1 Report
Nesfatin-1 is a satiety-inducing adipokine with a role in energy balance regulation, NUCB2 gene encodes a 396 amino acid, long precursor peptide, as well as a 24 amino acid long signal peptide. NUCB2/nesfatin-1 is abundantly expressed in several regions of the hypothalamus that control food intake. Indeed, it has been shown that NUCB2/nesfatin-1 reduces food intake in rodents when administrated either centrally or peripherally. Of note, NUCB2/nesfatin-1 is also widely distributed in various other tissues. This study was designed to expand knowledge on the localization of Nesfatin-1 in the digestive tract of an aquatic mammalian species, using the common bottlenose dolphin and allowing a comparative study on terrestrial mammals. Samples were derived from stranded animals or from marine mammals who died in captivity and were referred for post-mortem. The following samples analysed: first chamber, second chamber, third chamber or and intestine. The distribution of Nesf-1 was studied by immunohistochemistry on sections of the tissue’s samples.
They demonstrated the presence of Nesfatin-1 positive cells in the mucosal epithelium of both second and third chambers, and very interestingly absence in the intestine. In both rodents and human NUCB2/Nesfatin- 1 has been revealed in the middle and lower portions of gastric mucosal glands together in the submucosal layer of duodenum. These results show interesting and novel data on the distribution of Nesfatin-1 in the gastrointestinal tract of dolphins, considering the evolutionary position of these mammalian species. Interestingly showing some shared features between common bottlenose dolphins and terrestrial mammals are conserved.
The results are very interesting, however, there was no indication regarding the number of samples used per section (example n=), it would be interesting to also see that these finding are consistent when doing more bottlenose dolphins numbers. As the samples were coming from marine mammals who died in captivity was a much more ethical approach for sample collection.
The manuscript is clear and well written and bring a new light to nesfatin-1 distribution in other mammals. These novel finding highting an important area between evolution, feeding habits and digestive physiology. The results are clear and well represented and addressing the hypotheses well. The methodology is detailed and easy to follow as well as understand each step. With clear sections in all the figures and well written discussion and conclusion.
Author Response
Dear Editor,
we are grateful to the reviewers for their constructive comments and suggestions that enhance the article quality.
Below the list of point-by-point replies to the reviewer’s comments, which have been integrated in the main text and highlighted by using track changes.
Nesfatin-1 is a satiety-inducing adipokine with a role in energy balance regulation, NUCB2 gene encodes a 396 amino acid, long precursor peptide, as well as a 24 amino acid long signal peptide. NUCB2/nesfatin-1 is abundantly expressed in several regions of the hypothalamus that control food intake. Indeed, it has been shown that NUCB2/nesfatin-1 reduces food intake in rodents when administrated either centrally or peripherally. Of note, NUCB2/nesfatin-1 is also widely distributed in various other tissues. This study was designed to expand knowledge on the localization of Nesfatin-1 in the digestive tract of an aquatic mammalian species, using the common bottlenose dolphin and allowing a comparative study on terrestrial mammals. Samples were derived from stranded animals or from marine mammals who died in captivity and were referred for post-mortem. The following samples analysed: first chamber, second chamber, third chamber or and intestine. The distribution of Nesf-1 was studied by immunohistochemistry on sections of the tissue’s samples.
They demonstrated the presence of Nesfatin-1 positive cells in the mucosal epithelium of both second and third chambers, and very interestingly absence in the intestine. In both rodents and human NUCB2/Nesfatin- 1 has been revealed in the middle and lower portions of gastric mucosal glands together in the submucosal layer of duodenum. These results show interesting and novel data on the distribution of Nesfatin-1 in the gastrointestinal tract of dolphins, considering the evolutionary position of these mammalian species. Interestingly showing some shared features between common bottlenose dolphins and terrestrial mammals are conserved.
The results are very interesting, however, there was no indication regarding the number of samples used per section (example n=), it would be interesting to also see that these finding are consistent when doing more bottlenose dolphins numbers. As the samples were coming from marine mammals who died in captivity was a much more ethical approach for sample collection.
The manuscript is clear and well written and bring a new light to nesfatin-1 distribution in other mammals. These novel finding highting an important area between evolution, feeding habits and digestive physiology. The results are clear and well represented and addressing the hypotheses well. The methodology is detailed and easy to follow as well as understand each step. With clear sections in all the figures and well written discussion and conclusion.
In the paragraph “animals and sample preparations” we have specified the number of analyzed samples used for the analysis.

Reviewer 2 Report
The study is very well-conducted and nicely written.
This study is fascinating and has novelty. The conclusion of this study is very supportive of the data presented here. I have some comments, as mentioned below.
Comments:
- Introduction: Lines 46-49 needs a reference.
- Methods: it is not clear how many animals were used for these studies and there is not details about their age and metabolic state. As a satiety molecule, the expression of nesfatin-1 is regulated by different diets under different feeding regimes. Also, Nefatin-1 expression is different in different tissues.
- Results: Negative and positive staining images are not presented
- Image quantification will add value to the findings
- As Nesaftin-1 expression varies between different tissues, it will be interesting to see whether its expression patterns resemble human and murine models in other tissues such as liver, lungs, adipose tissue.
- The discussion will benefit from a section on limitations of the study. The fact that only one technique is used for measuring nesfatin-1 levels could be justified.

Author Response
Dear Editor,
we are grateful to the reviewers for their constructive comments and suggestions that enhance the article quality.
Below the list of point-by-point replies to the reviewer’s comments, which have been integrated in the main text and highlighted by using track changes.
The study is very well-conducted and nicely written.
This study is fascinating and has novelty. The conclusion of this study is very supportive of the data presented here. I have some comments, as mentioned below.
Comments:
- Introduction: Lines 46-49 needs a reference.
The reference was added
- Methods: it is not clear how many animals were used for these studies and there is not details about their age and metabolic state. As a satiety molecule, the expression of nesfatin-1 is regulated by different diets under different feeding regimes. Also, Nefatin-1 expression is different in different tissues.
For this study, we used samples collected from 3 adult males. At the time of the necropsy, the three chambers and intestine were rather empty, assuming that animals had not been fed over the last hours.
- Results: Negative and positive staining images are not presented
We added in the supplementary file images showing negative and positive controls (Figure S1).
- Image quantification will add value to the findings
All results were summarized in a table, which has been included in the main text. Majority of immunopositivity was observed in the glandular epithelium and in neurons and neuronal fibers of the ENS. An in-depth quantification of immunopositivity would be biased by the wider distribution in the mucosa and myenteric plexuses.
As Nesaftin-1 expression varies between different tissues, it will be interesting to see whether its expression patterns resemble human and murine models in other tissues such as liver, lungs, adipose tissue.
We thank you for the suggestion, which is the outlook of these preliminary data, presented to complement previous studies on the characterization of the neuropeptides in the gastro-entero-pancreatic system of marine mammals (Russo et al. 2012; Gatta et al. 2014; Gatta et al. 2018).
- The discussion will benefit from a section on limitations of the study. The fact that only one technique is used for measuring nesfatin-1 levels could be justified.
We are grateful for suggestion, and we have now improved the section of discussion including concerns of using bottlenose dolphins as experimental animals and thus limiting more detailed analysis through the use of multiple techniques.

Reviewer 3 Report
In this manuscript, the authors examined the expression of Nesf-1 in bottlenose dolphin digestive tract. The conclusion is solid with images of the gastric chambers using immuno staining method. The results are supported with the immunopositivity in nerve fibers and neurons. The study can add information to the current understanding of Nesf-1 in dolphin digestive system. The manuscript is suggested to be accepted in the journal.
Author Response
Dear Editor,
we are grateful to the reviewer endorsing and appreciating the manuscript.
In this manuscript, the authors examined the expression of Nesf-1 in bottlenose dolphin digestive tract. The conclusion is solid with images of the gastric chambers using immuno staining method. The results are supported with the immunopositivity in nerve fibers and neurons. The study can add information to the current understanding of Nesf-1 in dolphin digestive system. The manuscript is suggested to be accepted in the journal.

Reviewer 4 Report
The authors identified the anatomical distribution of nesfatin-1 immunoreactivity in the gastrointestinal tract of the common bottlenose dolphin. This is an interesting study, but some questions arise. Authors should take into consideration the following comments.
1. The author stated that they undertake this study to explore the pattern of distribution of nesfatin-1 to conduct further comparative studies with other mammals. Please further elaborate how these data allow further comparative study on terrestrial mammals.
2. The nesfatin-1 primary antibody was raised in rabbit. Did the authors verify the specificity of the primary antibody? What is the % similarity in the amino acid sequence between the bottlenose dolphin and rabbit nesfatin-1?
3. The unit of the scale bar on the microscopy images should be “µm” instead of “µ”.
4. Including western blot and RT-PCR data would add more useful information to the study and increase the overall merit of this article.
5. Figure 1 is not an original finding, please omit it from the results section or combine it with Figure 2 and 3 as a supplementary information of the anatomical location of the gastric chambers.
6. The samples were collected from how many animals?
7. Line 126, Positivity should be replaced with Positive cell or positive immunoreactivity.
8. There are minor discrepancy in the text and the citations, e.g. Ref 3 and 5.
9. Line 49, a reference is missing.
Author Response
Dear Editor,
we are grateful to the reviewers for their constructive comments and suggestions that enhance the article quality.
Below the list of point-by-point replies to the reviewer’s comments, which have been integrated in the main text and highlighted by using track changes.
The authors identified the anatomical distribution of nesfatin-1 immunoreactivity in the gastrointestinal tract of the common bottlenose dolphin. This is an interesting study, but some questions arise. Authors should take into consideration the following comments.
- The author stated that they undertake this study to explore the pattern of distribution of nesfatin-1 to conduct further comparative studies with other mammals. Please further elaborate how these data allow further comparative study on terrestrial mammals.
We have better highlighted the importance of further comparative studies : “Being cetaceans highly nested within the artiodactyl phylogenetic tree, it will be interested characterize Nesfatin-1 expression in the Ruminantia.”
- The nesfatin-1 primary antibody was raised in rabbit. Did the authors verify the specificity of the primary antibody? What is the % similarity in the amino acid sequence between the bottlenose dolphin and rabbit nesfatin-1?
The antibody recognizes the human N terminal: https://www.phoenixpeptide.com/products/view/Antibodies/H-003-24.
As written at line 188 there is a homology of 95% of the amino acid sequence between Tursiops truncatus and human Nesf-1 protein, including that epitope. Furthermore, this antibody was used already in a previous work on pancreas of bottlenose dolphin (Gatta et al. 2018 doi: 10.3389/fphys.2018.01845).
- The unit of the scale bar on the microscopy images should be “µm” instead of “µ”.
The scale bars were changed as requested.
- Including western blot and RT-PCR data would add more useful information to the study and increase the overall merit of this article.
All samples derived from stranded dead animals and were provided us by the MMBT, which mainly collects fixed and embedded tissues, making thus impossible to perform additional investigations such as western blot or RT-PCR.
- Figure 1 is not an original finding, please omit it from the results section or combine it with Figure 2 and 3 as a supplementary information of the anatomical location of the gastric chambers.
We moved the figure 1 in the supplementary materials, Figure 1 is now Figure S2.
- The samples were collected from how many animals?
For this study we used samples derived from 3 adult males.
- Line 126, Positivity should be replaced with Positive cell or positive immunoreactivity.
The word was changed as requested.
- There are minor discrepancy in the text and the citations, e.g. Ref 3 and 5.
The references were changed.
- Line 49, a reference is missing.
A reference was added.

Round 2
Reviewer 2 Report
The authors have addressed all the comments thoroughly and I am happy with the revised version.
Reviewer 4 Report
Thank you for the revised manuscript. The quality of the manuscript has improved.